# Fractional Factorial Design to Evaluate the Synthesis and Electrochemical Transfer Parameters of h-BN Coatings

**DOI:** 10.3390/nano13232992

**Published:** 2023-11-22

**Authors:** Helen Figueroa, Juliet Aristizabal, Elías Reinoso-Guerra, Bárbara Arce, María José Vargas-Straube, Dana Gentil, Cristian Ramírez, José Cordero, Nelson P. Barrera, Carolina Parra

**Affiliations:** 1Laboratorio de Nanobiomateriales, Departamento de Ingeniería Mecánica, Universidad Técnica Federico Santa María, Avenida España 1680, Valparaíso 2390123, Chile; helen.figueroa.14@sansano.usm.cl (H.F.); juliet.aristizabal@usm.cl (J.A.); elias.reinoso.12@sansano.usm.cl (E.R.-G.); barbara.arce.lopez@gmail.com (B.A.); maru.vs@gmail.com (M.J.V.-S.); dana.gentil@usm.cl (D.G.); 2Departamento de Ingeniería Química y Ambiental, Universidad Técnica Federico Santa María, Avenida España 1680, Valparaíso 2390123, Chile; cristian.ramirez@usm.cl; 3Departamento de Física, Universidad Técnica Federico Santa María, Avenida España 1680, Valparaíso 2390123, Chile; 4Centro Científico Tecnológico de Valparaíso, Universidad Técnica Federico Santa María, Calle General Bari 699, Valparaíso 2340000, Chile; 5Facultad de Ciencias Biológicas, Pontificia Universidad Católica de Chile, Alameda 340, Santiago 7820436, Chile; jocorderf@gmail.com (J.C.); nbarrera@bio.puc.cl (N.P.B.)

**Keywords:** h-BN, fractional experiment design, CVD synthesis, electrochemical transfer

## Abstract

In this study, we present a fractional factorial design approach for exploring the effects and interactions of key synthesis and electrochemical transfer parameters on the roughness and wettability of hexagonal boron nitride (h-BN) coatings, due to their essential role in biofilm formation. The studied parameters for the synthesis process include precursor mass, growth time, and substrate conditioning, whereas for the transfer process, applied voltage and aqueous medium concentration were studied. Through this polynomial model, we confirmed the strong influence of precursor mass and medium concentration parameters on h-BN surface roughness and its resulting antibiofilm properties.

## 1. Introduction

The formation of biofilm exerts substantial influence on the persistence of biofouling concerns within industrial and medical domains, since the microorganisms that live within a biofilm present increased resistance to cleaning agents, antibiotics, and environmental factors [1].

Within a biofilm, bacteria primarily reside in colonies embedded in a complex matrix of extracellular polymeric substances that confer mechanical stability and shield against environmental challenges, which also provide pathways for nutrients and molecular signals [2]. In a mature biofilm, the eradication of bacteria or the mechanical detachment of biofilms from solid surfaces becomes exceedingly challenging, and consequently, it becomes imperative to disrupt the initial stages of biofilm formation related to the sensing and binding of bacteria to surfaces.

It is noteworthy that the process of bacterial adhesion is intricately affected by a myriad of factors, including the surface properties of materials [3,4]. In recent decades, many research efforts have been devoted to understanding how surface charge [5], wettability [6], roughness [7], topography [8], stiffness [9], and their combinations influence bacterial adhesion [10,11,12,13]. More recently, innovations in nanotechnology-based systems have been proposed to eradicate bacterial biofilms [14], such as using controlled nanostructured textures [15,16], mimicking biological surfaces like lotus leaves, shark skin, or mollusk shells [17,18,19], and using nanomaterials as surface coatings [20,21,22,23]. Among these last surface modification approaches, we found novel strategies such as nanoparticle-based composites that have been explored for treating multidrug-resistant bacteria [24], coatings containing graphene oxide for controlling biofilm formation on inverse osmosis membranes [25], and silver nanoparticle-based coatings whose antifouling activity has been tested using urinary pathogens [26].

Some bidimensional (2D) nanostructured materials have also shown relevant antibacterial mechanisms, including direct contact destruction, oxidative stress, and photo-induced antibacterial mechanisms [27]. Graphene [28,29,30,31,32], and molybdenum disulfide [33] are two examples of low-dimensional antibiofilm coatings that have been reported to exhibit complex and controversial interactions with biological systems [34]. Recent advancements in the bulk synthesis of 2D nanomaterials have opened the possibility of using them as another surface modification pathway for antibiofilm purposes [35,36,37]. Within these 2D materials, hexagonal boron nitride (h-BN) has been reported to present antibacterial and antibiofilm performance in nanoparticle form [38], such as flakes or nanotubes [38,39,40,41,42,43,44]. However, 2D h-BN coatings for controlling biofilm formation and proliferation are still rather unexplored [34,45].

In general, the tuning of coatings’ specific surface characteristics will allow for the engineering of the desired antibiofilm functionalities. For instance, the physicochemical characteristics of substratum materials strongly impact the early interaction between material and bacterial system [46]. Among them, surface roughness has been identified as an essential factor affecting cellular attachment to surfaces prior to biofilm formation [13,47] and the long-term development of biofilms [48]. In general, at higher surface roughness, bacterial adhesion rises, but at a certain limit, it can start declining again, an effect that is connected to the reduced contact area between the nanostructured surface and the bacteria. In addition, the combined effect of roughness and surface energy plays a main role in determining surface wettability, which has been proven to have a dramatic impact on bacterial attachment and how quickly biofilms develop [49]. Surface wettability is characterized by the contact angle of the surface, where the hydrophilic/phobic nature of a microorganism interplays with the wettability of the surface [50], and a moderate wettability (water contact angle close to 90°) leads to higher bacterial adhesion, and superhydrophilic and superhydrophobic surfaces lead to lessened bacterial adhesion [51].

A multitude of factors come into play when tuning the surface characteristics of 2D coatings through the synthesis process. Single-layer and few-layer hexagonal boron nitride (h-BN) coatings can be obtained by means of the chemical vapor deposition process (CVD), and its subsequent transfer. The relevant factors of this synthesis process are growth temperature, catalytic substrate, and precursor quantity, among others, whereas for an electrochemical route of the transfer process, the delamination applied voltage and the solution concentration are the tunable variables.

Synthesis and transfer process variables assume distinct roles that collectively shape the final characteristics of the h-BN coating [52]: growth substrate crystallinity [53] can guide the crystalline orientation of resulting h-BN, precursor nature [54] and synthesis time [55,56] can influence the film’s final thickness, and the growth temperatures of post-growth thermal treatments can impact the crystallinity of the film [57]. Abundant research has been conducted regarding the controllable synthesis of high-quality h-BN crystals and thin films with desirable properties by studying each synthesis and transfer process variable at a time [58,59]. However, this approach has limitations, including the inability to assess interactive effects between the variables, which can be labor-intensive and inefficient using traditional approaches. In this study, we embark for the first time on a fractional analysis design approach to delve into the role of multiple h-BN CVD synthesis and transfer variables, with the overarching objective of achieving the desired match between surface roughness and wettability, to unlock its potential for effective antibiofilm applications.

Factorial design has often been used for the control of the synthesis process of nanomaterials such as nanoparticles [60,61,62], nanocomposites [62], and nanofibers [63], but has not been used before for CVD growth of h-BN. This approach not only allows us to determine which factors are relevant (or not) when synthesizing this material, but these experiments also give us a clearer idea of the degree to which each variable interacts with each other. This type of experimental design is an important tool for making a quick evaluation of different responses that depend on several factors, both for industry and for scientific work [64]. Furthermore, factorial design is the first approach to knowing the importance of and interaction between each variable. After this analysis, more experiments can be carried out with the factors found most significant.

The resulting h-BN samples were tested as an antibiofilm coating using *E. coli*, which is a significant source of medical device-related infections when it develops into a biofilm [65].

## 2. Material and Methods

### 2.1. Synthesis of h-BN

Hexagonal boron nitride (h-BN) was synthesized via a chemical vapor deposition (CVD) method at atmospheric pressure using amino borane BH_6_N (Sigma Aldrich, St. Louis, MO, USA, 99% purity) as a precursor and a 25 mm thick copper foil (Alpha Aesar, Haverhill, MA, USA, 99.999% purity) as substrate. Two different conditions for the starting copper substrate were used: a recently acquired foil (“pristine”), and one that has been storage for 5 years and showed evident signs of surface oxidation (“oxidized”). See Appendix A for details. All copper substrates (oxidized and pristine) were cut into pieces of 6.0 × 8.0 mm, pretreated via heating with a hot plate at 250 °C to oxidize the surface, and then immersed in a bath of 10% nitric acid for 45 s to remove the oxide layer. This sheet was taken in a quartz tube of 60 mm diameter and placed at the center of a furnace heating area for further h-BN growth process.

The temperature was then raised to 1030 °C under an atmosphere composed of a mixture of argon and hydrogen gas at a flow rate of 50 sccm and 20 sccm, respectively [66,67]. The furnace was equipped with a pre-chamber for the thermal decomposition of the precursor (at a temperature between 90 °C) [68], with the gas flow diverted into the chamber before entering the furnace [68]. When 1030 °C was reached, the gas flow was driven to the pre-chamber to carry the synthesis precursor gases produced in the decomposition towards the furnace chamber. A quartz wool filter was placed at the exit from the pre-chamber to the furnace to prevent solid particle formation in the furnace during precursor decomposition (Appendix A).

### 2.2. Electrochemical Transfer Process

The electrochemical delamination method was used for transferring h-BN into a SiO_2_ substrate (Figure 1). First, a thin layer of PMMA (polymethyl methacrylate, Sigma Aldrich, 350,000 molecular weight) was deposited through spin-coating at 2000 rpm on top of h-BN grown on copper, and heated to 180 °C for 2 min to cure the PMMA film. The PMMA-coated sample was then placed in nitric acid, with the uncoated side facing the solution for 30 s, and further washed with deionized water to remove acid residues. A potentiostat was used to electrochemically delaminate the PMMA/h-BN layer from Cu, using a sodium hydroxide (NaOH) solution, the PMMA/h-BN/Cu substrate as a negative electrode, a platinum wire electrode as a positive electrode, and a reference electrode of Ag^+^/AgCl. The detached PMMA/h-BN layer was washed repeatedly in deionized water before transferring it into a SiO_2_ substrate. After transferred, the sample was dried for 12 h and then heated to 185 °C for 2 min to minimize h-BN wrinkles. Finally, the PMMA layer was removed from the h-BN/SiO_2_ sample with successive washes of acetone and isopropanol.

### 2.3. Characterization

#### 2.3.1. Optical Microscopy

A Nikon Eclipse LV100ND microscope (Nikon, Tokyo, Japan) was used to explore the coverage of the h-BN/Cu samples via optical microscopy. An oxidation in air procedure was performed at 200 °C on the h-BN grown on copper samples to improve the optical contrast of the coated and uncoated areas [69] (Appendix A).

#### 2.3.2. RAMAN Spectroscopy

Raman analysis was performed on h-BN samples using a modular Micro Raman confocal spectrometer system (Renishaw, 532 nm laser, Gloucestershire, UK), using a visible excitation laser at 532 nm.

#### 2.3.3. Surface Roughness

The morphology and roughness of h-BN/SiO_2_ samples were studied using atomic force microscopy MFP-3D-SA (Asylum Research, Santa Barbara, CA, USA) in tapping mode. The parameters that were quantified to analyze surface roughness were the root mean square roughness (RMS) and the roughness ratio. The RMS roughness was calculated based on the root-mean-square of the height of microscale peaks and valleys as a means of quantifying the average feature size [70]. The roughness ratio was calculated by dividing the actual surface area by the projected area. Autocorrelation length was obtained from the analysis of the power spectral density function (PSDT). WSxM 4.0 Beta 9.3 software [71] was used to analyze the AFM images and calculate the afore mentioned parameter values for each sample.

#### 2.3.4. Contact Angle

The wettability of the h-BN/SiO_2_ samples was measured using the contact angle. A 5 µL drop of distilled water was placed on the sample surface, and images were immediately captured using a high-resolution camera (Navitar Zoom 7000 Navitar TV Zoom Japan Camera, Navitar, Denville, NJ, USA). The contact angle was measured using Image J software (1.54d-win-java8) with the Drop Shape Analysis add-on (“drop analysis” LB ADSA-https://doi.org/10.1016/j.colsurfa.2010.04.040)

#### 2.3.5. Morphology

Scanning electron microscopy (SEM) images of h-BN samples were obtained using a Carl Zeiss microscope EVO MA-10 (Carl Zeiss Pty Ltd., Macquarie Park, New South Wales).

### 2.4. Preparation of Bacterial Cultures

For the assessment of bacterial biofilm formation on the h-BN/SiO_2_ samples, *Escherichia coli* K-12 MG1655 was used. Luria Bertani medium (Tryptone (10 g/L), yeast extract (5 g/L) and NaCl (10 g/L)) was inoculated with an isolated colony of the strain and incubated overnight at 30 °C and 150 rpm. The inoculum was adjusted to an optical density of 0.1 (at OD600) and cultivated for 6 more hours using the same conditions. Subsequently, the culture was harvested, washed with a saline phosphate buffer (PBS), dissolved in deionized sterile water, adjusted again to a 1.5 optical density (OD600), and an initial count of the colony-forming units (CFU) was performed. Droplets of 300 μL of inoculum were deposited over the h-BN/SiO_2_, and the uncoated SiO_2_ control samples and were incubated for 24 h at 30 °C without stirring.

Finally, the samples were harvested, the planktonic phase was discarded from the cultures, and the samples were washed with distilled water and PBS for their analysis with microscopy.

Scanning electron microscopy (SEM) was performed to evaluate bacterial adherence to the samples and biofilm architecture. For sample preparation, samples were fixed in 3% (*v*/*v*) glutaraldehyde for 48 h, dehydrated via serial washing with ethanol at concentrations from 10% to 100% *v*/*v*, critical point drying, and gold coating, and kept in a desiccator until analysis via SEM. The analysis was performed using an ESEN ThermoScientific (Waltham, MA, USA) Quattro S field emission scanning electron microscope with ETD detection and 30,000 kV voltage. Initially, the samples were observed using 500× magnification to visualize the entire surface and assess the degree of homogeneity of bacterial colonization. Then, several representative areas of the entire surface, both in the center and at the edge of the samples, were chosen and observed using higher magnification, from 2000× to 10,000×, to evaluate the architecture of the biofilms. Finally, the chosen areas were observed at 25,000× and 50,000× magnifications to evaluate the cellular morphology of the attached bacteria.

### 2.5. Fractional Factorial Experimental Design for Quality of h-BN Transferred Layer

Fractional factorial is a technique for experiment design used to estimate the effect of each variable of a process on a specific response, with a reduced number of experiments when compared to the traditional approaches that study the effect of one factor at a time. In the case of three factors, the data fit a model that takes the following form:(1)$$y=I+a_{1}\cdot A+a_{2}\cdot B+a_{3}\cdot C+a_{12}\cdot AB+a_{13}\cdot AC+a_{23}\cdot BC+a_{123}\cdot ABC\;$$

To understand the effects of the synthesis and transfer process parameters on the roughness and wettability of h-BN, three synthesis variables and two transfer process variables that are determinants in the final quality of the substrate were selected [72,73,74,75,76,77]. The synthesis variables are the amount of precursor (A), the condition of the copper substrate (B), and synthesis time (C), whereas the selected transfer variables are the transfer voltage (D) and the NaOH electrolyte concentration (E). A $2_{III}^{5-2}$ fractional factorial design resolution III was used to investigate the quality performance of the h-BN/SiO_2_ samples by measuring their roughness and contact angle. The factors and the selected lower and upper values and its values are as follows: A—amount of precursor (11 and 22 mg) [74,78], B—condition of the copper substrate (oxidized and pristine), C—synthesis time (30 and 60 min), D—transfer voltage (1 and 4 V), and E—Sodium hydroxide (NaOH) concentration (0.01 and 0.10 M).

The choice to use fractional factorial was made for practical reasons and cost minimization; to fully model five factors in a full factorial design, 32 ($2^{5}$) independent experiments are necessary, without considering their due replicates; in this fractional factorial model, we will use only eight experiments. Each sample is denoted as S*_i_*, where i (from 1 to 8) corresponds to each specific experimental condition, as follows in Table 1

To carry out the fractional factorial design, a subset of the complete factorial design must be chosen. For this, we decided to confuse two variables. The generating basis of the subset is obtained by confounding the effect of the variables D = −AC and E = ABC, under the assumption that the main effects are more important than the second and third-order effects, generating the alias structure shown in Table 2. The alias structure describes the pattern of confusion that occurs in a design; the confused terms then are said to be aliased.

## 3. Results and Discussion

### 3.1. Characterization

For the Raman characterization, the h-BN/SiO_2_ samples were analyzed using a 532 nm excitation wavelength laser (Figure 2). All samples show an absorption band centered around 1370 cm^−1^, which is consistent with the h-BN characteristic signal, although in some samples, the peak is blue-shifted because of the hardening of E_2G_ phonon due to shorter B-N bonds presumably connected to compressive stress caused by the stretching of the film [79,80].

AFM imaging of the h-BN samples (Figure 3) allows us to obtain the RMS roughness of the 2D coatings (Figure 4), whose values are between 1 nm and 3 nm. The samples S_3_, S_5_, and S_7_, which show some granular areas, present the highest RMS values and dispersion (see Appendix A for complementary optical microscopy images). However, it is worth noting that these roughness characteristics are consistent with h-BN samples approaching few-layer thickness, as reported in previous studies [81,82].

The results of the contact angle measurements are shown in Figure 5 and Figure 6 Each of these measurements was performed in triplicate. Of the total samples, only S_2_ and S_6_ presented a contact angle larger than 90°, which indicates that their surface exhibits hydrophobic behavior. This may be due to the adsorption of organic molecules during their exposure to air. However, the mechanism of the change in the hydrophobic to hydrophilic nature of the h-BN coatings is not yet clearly understood. Interaction with water, on the contrary, may generate hydroxylation of the edges, enhancing the hydrophilic character of h-BN [83]. In terms of its application as a coating, it is important to note that the surface must combine its wettability and roughness to minimize bacterial adhesion, as hydrophobic and hydrophilic surfaces have been found to exhibit inverse adhesion tendencies [84], except in the range of 0.01–0.001 µm surface roughness, where materials have been found to exhibit no bacterial adhesion independent of their hydrophobic or hydrophilic nature [12].

### 3.2. Experimental Design

Table 3 summarizes the average RMS roughness and contact angle values for each factor across the two levels.

#### 3.2.1. Empirical Model of Roughness Equation

The analysis of variance (ANOVA) including all five factors and all possible interaction terms between them yielded the coefficient estimates shown in Table 4.

The RMS roughness responses observed in the 2^(5-2)^ fractional factorial design were used to conduct a regression analysis (Equation (2)) in uncoded units with square R adjustment 80.99%
(2)RMS=1.236−0.0195·A−0.331·B+0.0289·C−0.0070·D−6.99·E−0.0247·AB−0.047·AE

To evaluate the contribution of each parameter, the significance level (α=0.05) needs to be compared to the *p*-value (the probability of obtaining test results at least as extreme as the result observed). If α< *p*-value, the parameter’s contribution is rejected, and it is considered not significant. Appendix A shows the *p*-values for the parameters in Equation (2). The factorial analysis shows that the effects that are significant for h-BN roughness are the first-order effects of time, NaOH concentration (E), and amount of precursor (A); and the second-order effect, which considers the interaction between the amount of precursor and the type of substrate (AB) used in the synthesis (Appendix A). The alias structure analysis (Table 2) confirms a low contribution to the model of the second-order terms (4.44%), whereas a 79.38% contribution was found for the first-order terms.

Considering only the significant parameters, the model is expressed as follows:(3)RMS=1.236−0.0195·A +0.0289·C −6.99·E−0.0247·AB

Based on the model of Equation (3), contour plots (Appendix A) allow us to visualize the behavior of the roughness and the dependence between the variables, in the range of action of the factorial design. The non-parallel lines in the E (NaOH concentration) vs. C (Synthesis time) interaction plot indicate that the effect of factor E depends on the level of factor C, and to a lesser extent, similar results hold for the interactions of E vs. A (amount of precursor) and C vs. A. The model allows for the calculation of the combinations of factors that lead to maximum and minimum values for the response variable. Table 5 shows the resulting values.

#### 3.2.2. Empirical Model for Contact Angle Equation

The analysis of variance (ANOVA) including all five factors and all possible interaction terms between them yielded the coefficient estimates shown in Table 6.

The contact angle responses observed in Table 3 of the 2^(5-2)^ fractional factorial design [85] were used to conduct a regression analysis in uncoded units with square R adjustment 98.64%. Appendix A shows the *p*-values for the parameters in Equation (4).
(4)Contact Angle=65.07−2.109·A−5.82·B+−0.4174·C−0.233·D−714.5·E−0.1882·AB−42.31·AE

According to *p*-value analysis, the parameters that are significant for contact angle are the first-order effects of time, condition of the copper substrate (B), synthesis time (C) and amount of precursor (A), whereas the second-order effect considers the interaction between the amount of precursor (A) and the NaOH concentration (E).

In the case of the regression carried out for the contact angle, it was found that one of the parameters that had the greatest contribution (47.31%) is a second-order parameter that corresponds to the AE parameter; given that a fractional factorial design was carried out, this parameter also confuses other effects, as shown by the alias structure. This coefficient includes another second-order interaction BC and two third-order interactions ABD and CDE, given their non-significant contribution. Considering only the significant parameters, the model is expressed as
(5)Contact Angle=65.07−2.109·A−5.82·B+−0.4174·C−0.1882·AB−42.31·AE

Based on the model of Equation (5), in the range of action of the fractional factorial, the maximum contact angle should be 109.5°, resulting from a combination of parameters that is covered by the executed experiments. The minimum value is calculated to be 55.1°, and the conditions that would give this value land in another block of fractional factorial experiments. A summary of the values for parameters that give maximum and minimum roughness according to the model can be seen in Table 7.

#### 3.2.3. Effect of Amount of Precursor

Figure 7 shows roughness as a function of the quantity of precursor, where an increase in precursor quantity is accompanied by a decrease in roughness for all samples, except S3 and S6. In those cases, a key factor in the observed increase in roughness is the interaction of second-order effects between precursor quantity and substrate type. According to the theories outlined in earlier references [68,86,87], this phenomenon is probably linked to the growth of nucleation sites on the copper substrate. To analyze and elucidate the overarching trend of decreasing roughness with increased precursor quantity, one must consider the three stages involved in the h-BN growth mechanism:Precursor decomposition: during this phase, precursors like ammonia borane and borazine undergo breakdown into boron- and nitrogen-based compounds at elevated temperatures.Deposition and nucleation: The resulting boron- and nitrogen-based compounds are deposited onto metal surfaces, subsequently forming clusters.Continuous growth: These clusters then expand, forming larger h-BN islands, which eventually merge to create a seamless film.

Once this continuous film is established, the formation of new layers starts. Thus, a higher precursor volume leads to quicker coverage of the copper surface within a shorter period. However, extended exposure times result in the formation of new islands over this base layer [88]. These roughness results are in agreement with the observed Raman signals, where samples with the highest roughness exhibit signals below, indicating the presence of a few layers [79,89].

#### 3.2.4. Effect of the Copper Substrate Condition

Figure 8 displays the dependence of roughness on the condition of the Cu substrate (pristine or oxidized), and it is not possible to establish a clear dependence, indicating that pretreatment of copper is enough to standardize the surface for both base conditions and allow homogeneous nucleation of h-BN on the surface, even when low-quality copper is used as substrate [54,82,90].

#### 3.2.5. Effect of Synthesis Time

Figure 9 shows the effect of synthesis time (30 and 60 min) on h-BN roughness. In general, growth time increases the surface roughness [67], coinciding with the factorial analysis that indicates this is a positive effect parameter. This suggests that because more hexagonal boron nitride is deposited over time on the copper substrate, new islands are formed on top of other h-BN grain islands. However, the increase in roughness is not significant because, as reported by Auwärter [90], once the first monolayer is completed, the rate of h-BN formation decreases, since the precursor–metal interaction is favored over precursor–h-BN interaction. This makes the growth of a single-layer h-BN on metals easier than h-BN multilayer growth, which is why although an increase in roughness is observed, it remains in the range between 1 and 3 nm (single- and few-layer).

Furthermore, another factor that contributes to the increase of surface roughness over time is related to the thermal decomposition of the precursor over time and the species formed during the growth process [68]. At 30 min during the thermal decomposition of amino borane, three species are available: ammonia, amino diborane, and diborane. The dominant species during the first 30 min of synthesis is diborane, with its concentration peaking around 30 min before starting to decrease; however, the other species are still increasing in concentration, so that at 60 min of synthesis, a high amount of ammonia, amino diborane, triborane, and aminoborane are available, leading to a higher availability of B and N. This diminishes the Gibbs free energy barrier; thus, the smooth Cu surface effectively reduces the kinetic diffusion barrier and enhances the surface mobility of the free BN radicals that have longer diffusion lengths and move freely along the active edges, facilitating heterogeneous nucleation of h-BN on the surface with higher roughness [91].

#### 3.2.6. Effect of Transference Parameters

Different voltages and concentrations have been reported in electrochemical delamination transfer processes, but not in a comparative manner [76,77,92,93]. According to our results, the voltage does not seem to have a relevant effect on the roughness of transfer h-BN coatings, but the electrolyte concentration does.

Figure 10 shows the lack of a clear tendency of the influence of the applied voltage on the h-BN surface roughness, coinciding with the negative effect parameter found in the factorial analysis.

In terms of the concentration of electrolyte (NaOH), Figure 11 shows that smaller roughness can be found at higher concentrations of electrolyte used in the transference process. In general, at higher concentrations of NaOH, there is more conductivity and a larger delamination speed. This higher concentration of NaOH gives a total amount of Na^+^ (several orders of magnitude) that is high enough to provide an ionic screening effect against H^+^. On the other hand, this Na+ acts as an insulator shield and driving force of homogeneous bubbling between h-BN and Cu [94,95,96].

### 3.3. Bacterial Adhesion

To test the antibiofilm formation capability of the prepared h-BN samples, a bacterial adhesion analysis using *E. coli* cells was performed. These results confirm the suppression of biofilm formation on all h-BN coated samples (Figure 12), in contrast to uncoated SiO_2_ samples which present a three-dimensional architecture with the characteristic bacillary morphology of the strain.

The roughness values obtained for all h-BN coatings are in the range 0.1–6 nm, where bacterial adhesion is suppressed [12], consistent with results found in this study for the *E.coli* MG1655 strain. When observing the different zones of each sample at a higher amplification, it was determined that in some samples, there is the presence of bacillary bacteria adhered in isolation and not interacting with other bacteria, ruling out the possibility of biofilm formation in these samples. It is important to note that regardless of the hydrophilic or hydrophobic nature of the surfaces, we have observed no bacterial adhesion for all the h-BN coated samples.

## 4. Conclusions

In conclusion, our study aimed to explore the influence of synthesis and transference parameters on the roughness and contact angle of APCVD h-BN. To assess the interplay of CVD process parameters, we employed a fractional factorial for experimental design. Our findings revealed that concerning roughness, the most influential factors were time, NaOH concentration, and the amount of precursor. Additionally, the second-order effect, which considers the interaction between the amount of precursor and the type of substrate, played a significant role in shaping roughness.

In contrast, for the wettability property, the most notable interaction effect was observed in the second-order AE parameter corresponding to the interaction of precursor amount and NaOH concentration. It is important to note that in a fractional factorial design, this parameter can also confuse other effects, as demonstrated by the alias structure. This coefficient encompasses an additional second-order interaction BC and two third-order interactions ABD and CDE, which, up to this point, we consider non-significant. Due to their non-significant contributions, it becomes necessary to either conduct further fractional factorial experiments to disentangle these effects or introduce additional experiments that specifically isolate and identify the individual contributions of each parameter. According to the model, it is possible to modulate surface roughness within the range of 0.63 nm to 2.73 nm by changing the synthesis and transfer parameters. This low roughness, coupled with the distinctive physical and chemical properties of h-BN, including the electronegativity of the nitrogen bonds on the surface of hexagonal boron nitride, could potentially hinder bacteria from employing their typical mechanisms of interaction with surface atoms for colonization, thereby reducing bacterial adhesion.

## Figures and Tables

**Figure 1 nanomaterials-13-02992-f001:**
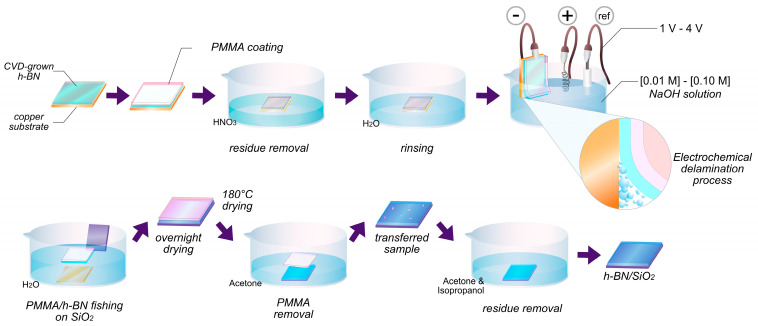
Diagram of the electrochemical transfer process of h-BN onto SiO_2_.

**Figure 2 nanomaterials-13-02992-f002:**
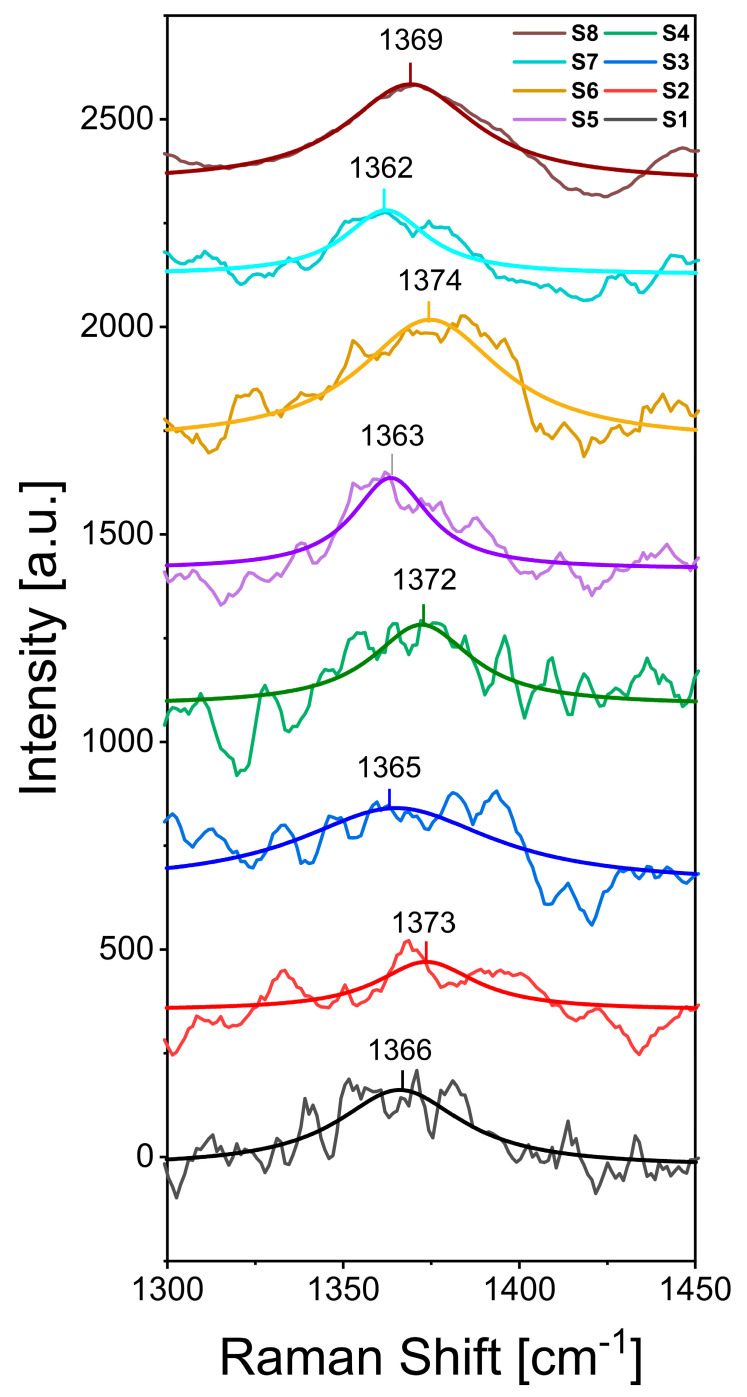
Representative Raman spectra of the h-BN/SiO_2_ samples obtained using conditions described in Table 1.

**Figure 3 nanomaterials-13-02992-f003:**
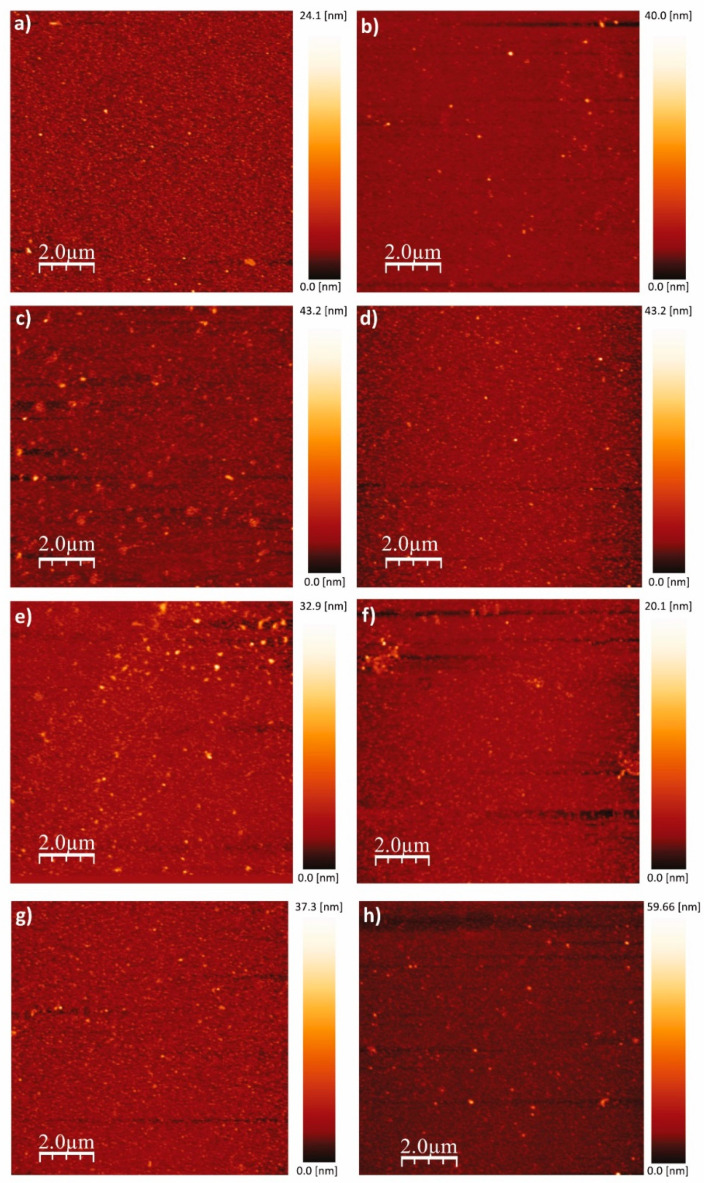
Representative AFM images of the h-BN/SiO_2_ samples. (**a**) S_1_, (**b**) S_2_, (**c**) S_3_, (**d**) S_4_, (**e**) S_5_, (**f**) S_6_, (**g**) S_7_, and (**h**) S_8_. All images are 10 µm × 10 µm, 256 × 256 points, 1 V, 1.3 kHz.

**Figure 4 nanomaterials-13-02992-f004:**
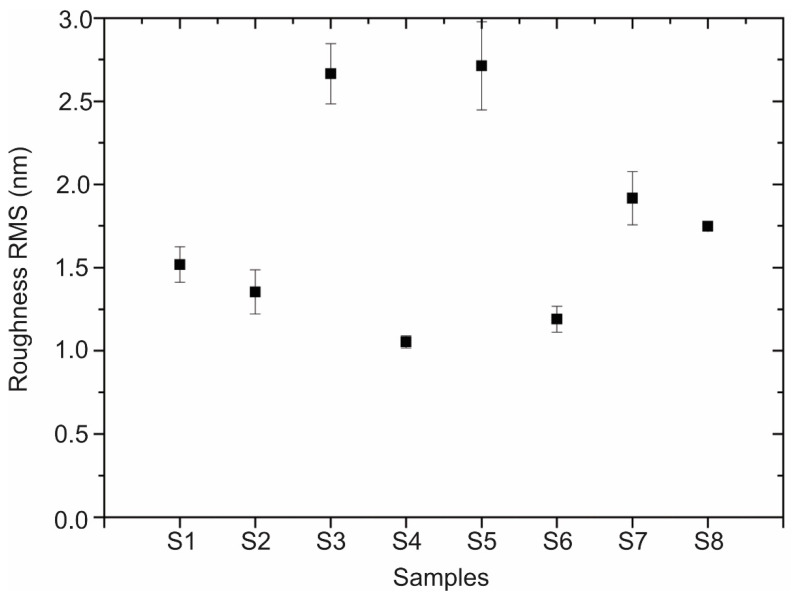
RMS roughness values of h-BN/SiO_2_ for each sample preparation condition.

**Figure 5 nanomaterials-13-02992-f005:**
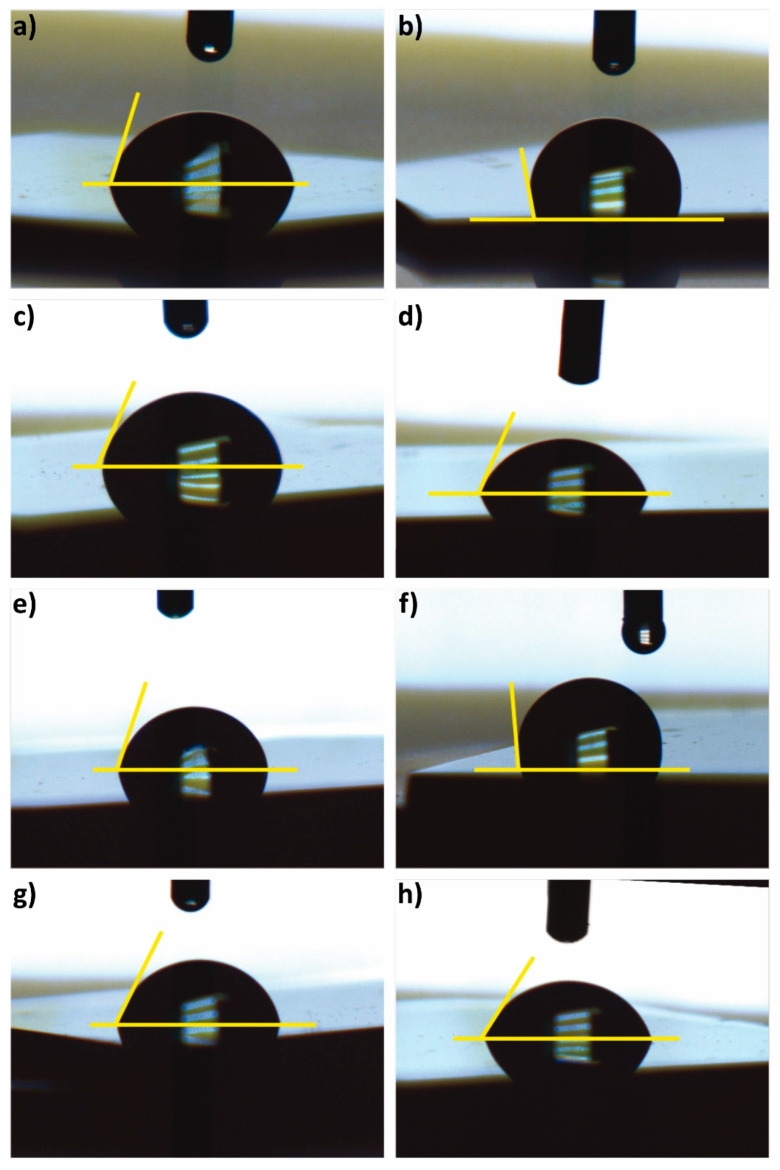
Contact angle images of h-BN/SiO_2_ samples: (**a**) S_1_, (**b**) S_2_, (**c**) S_3_, (**d**) S_4_, (**e**) S_5_, (**f**) S_6_, (**g**) S_7_, and (**h**) S_8_.

**Figure 6 nanomaterials-13-02992-f006:**
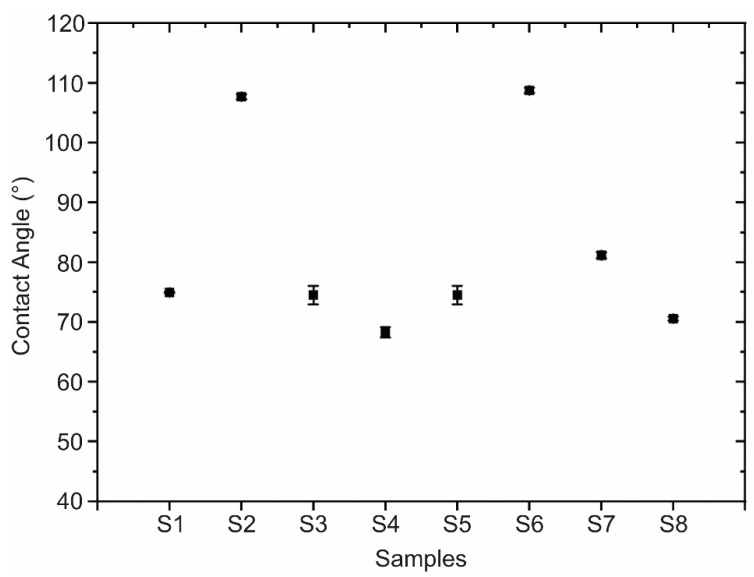
Contact angle values of each h-BN/SiO_2_ sample.

**Figure 7 nanomaterials-13-02992-f007:**
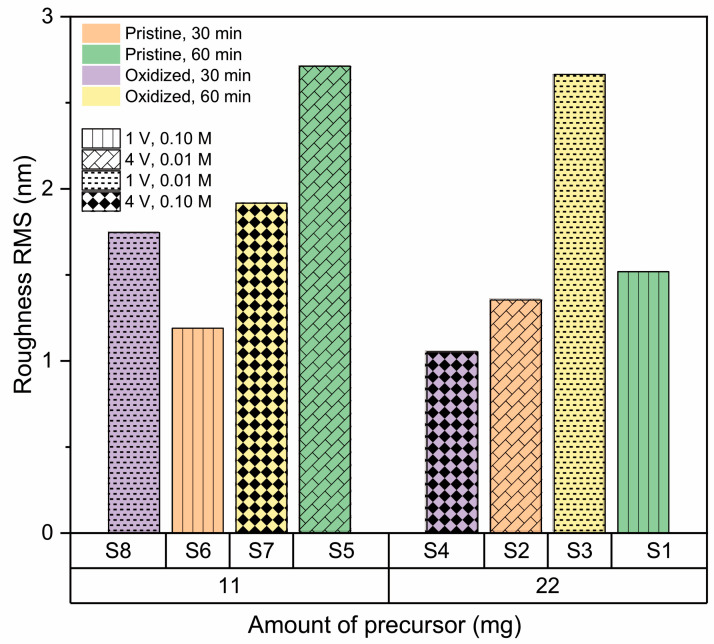
Effect of amount of precursor on roughness of h-BN/SiO2 samples for 11 mg and 22 mg.

**Figure 8 nanomaterials-13-02992-f008:**
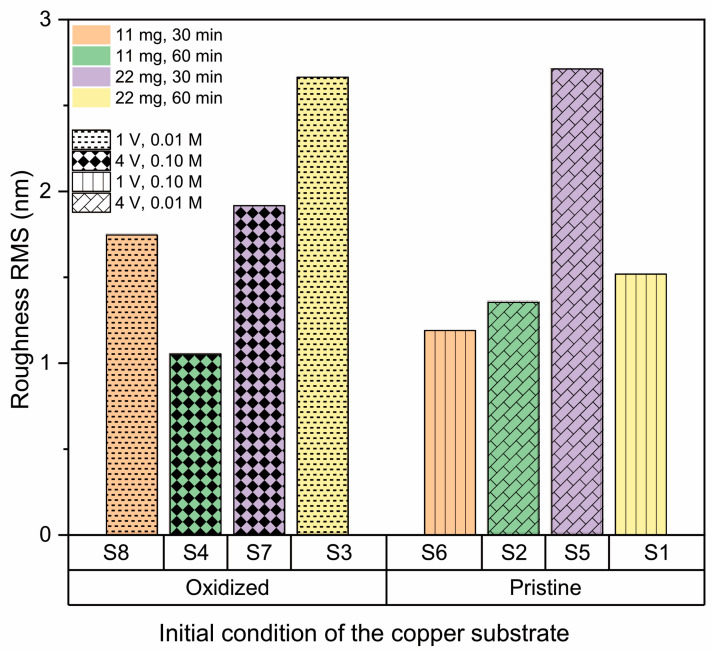
Effect of the condition of the copper substrate on the RMS roughness of h-BN/SiO_2_ samples for oxidized and pristine Cu.

**Figure 9 nanomaterials-13-02992-f009:**
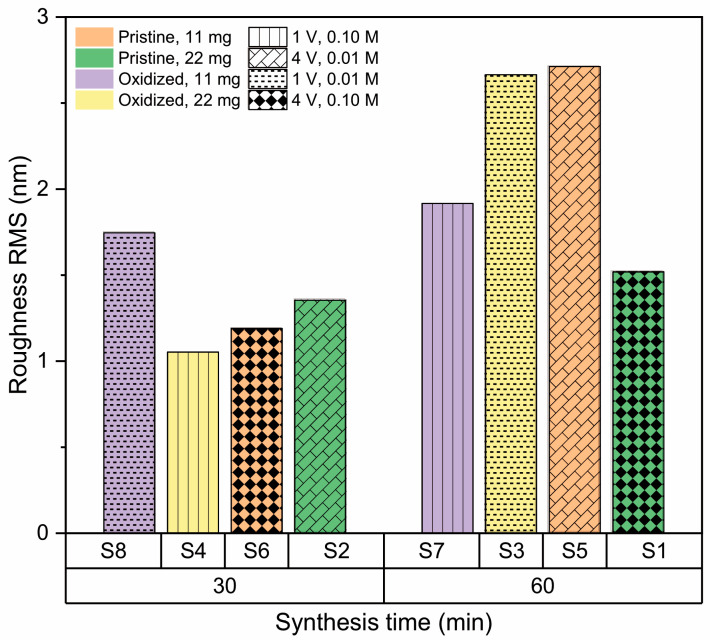
Effect of synthesis time on roughness of h-BN/SiO_2_ samples for 30 min and 60 min.

**Figure 10 nanomaterials-13-02992-f010:**
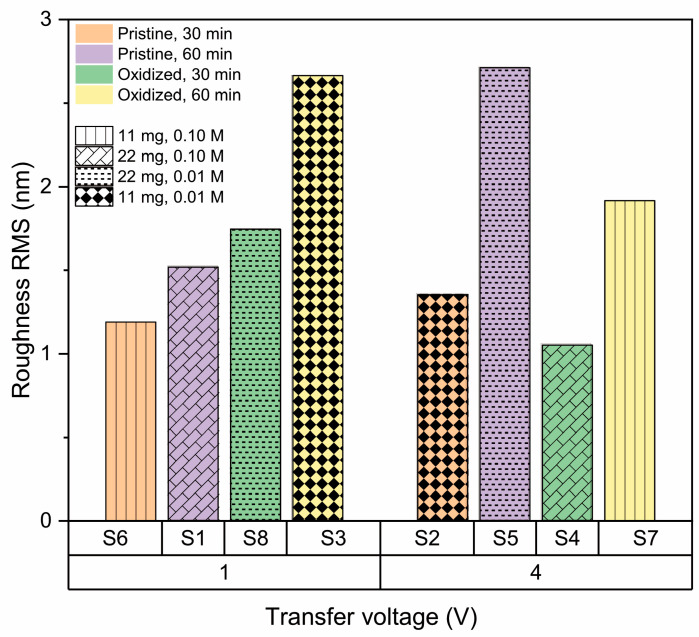
Effect of transfer voltage on the roughness of h-BN/SiO_2_ samples for 1 V and 4 V.

**Figure 11 nanomaterials-13-02992-f011:**
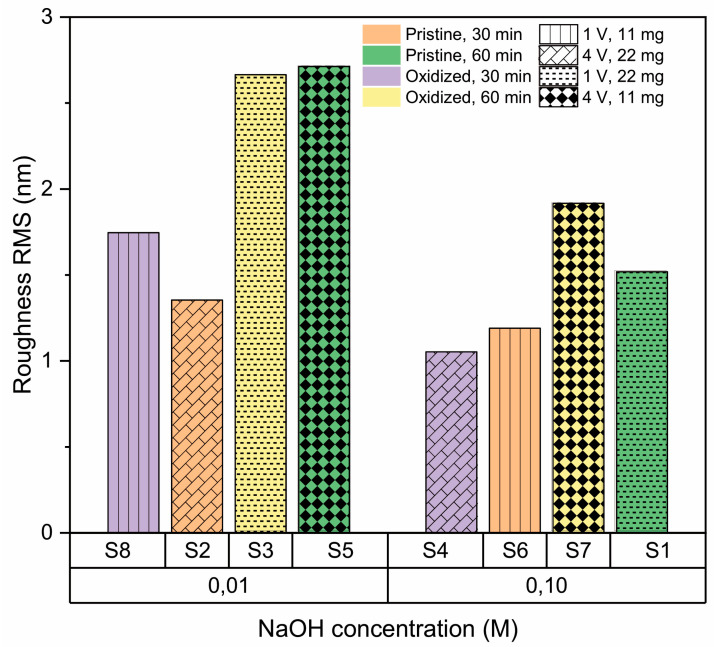
Effect of NaOH concentration on the roughness of h-BN/SiO_2_ samples for 0.01 M, and 0.10 M.

**Figure 12 nanomaterials-13-02992-f012:**
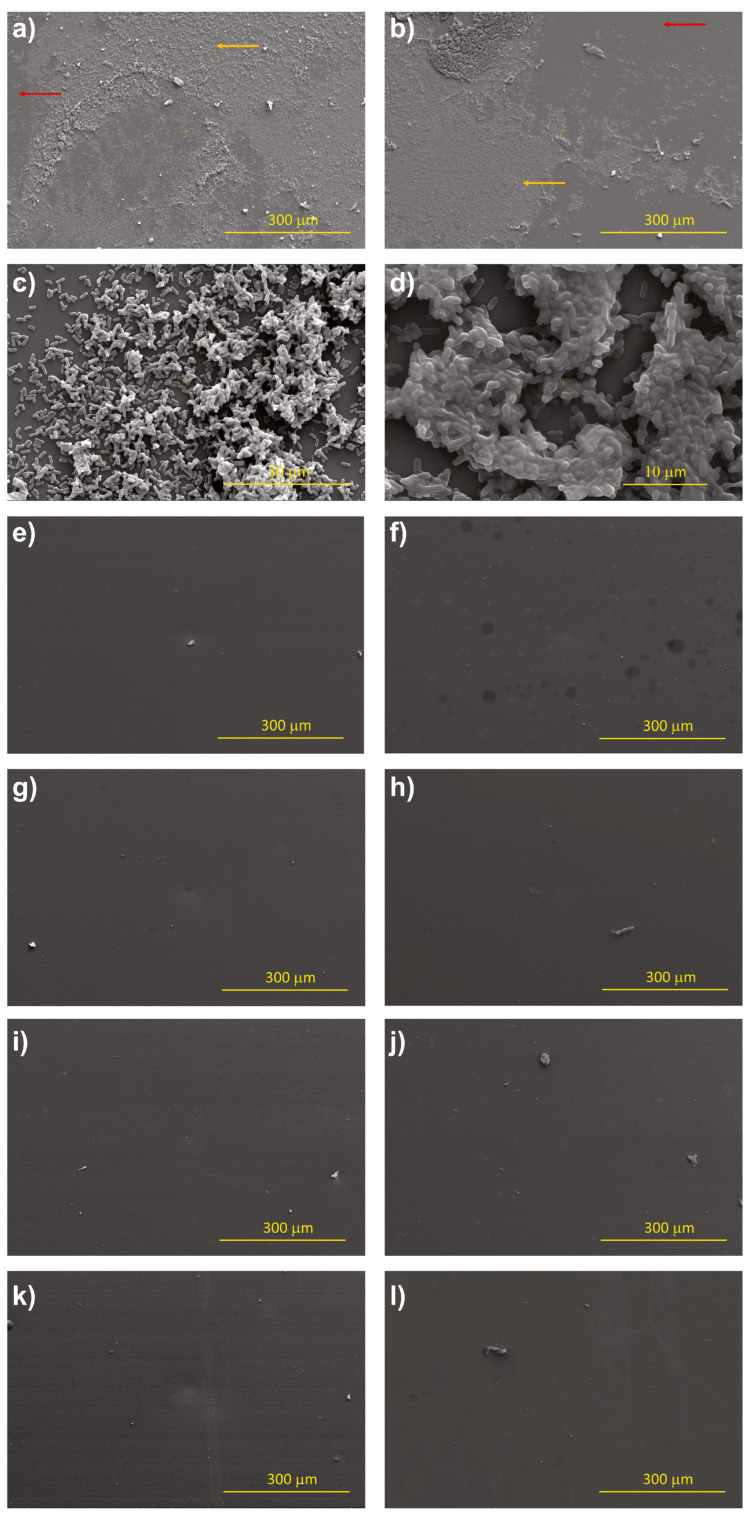
SEM images for adhesion of *E. coli* MG1655 on the h-BN coated and uncoated samples. (**a**,**b**): uncoated SiO_2_ at 500× magnification; (**c**,**d**): uncoated SiO_2_ at 10,000× magnification. H-BN/SiO_2_ samples. (**e**) S_1_, (**f**) S_2_, (**g**) S_3_, (**h**) S_4_, (**i**) S_5_, (**j**) S_6_, (**k**) S_7_, and (**l**) S_8_: all obtained at a magnification of 500×.

**Table 1 nanomaterials-13-02992-t001:** Experimental conditions for each sample’s preparation in fractional factorial.

Sample	A—Amount of Precursor (mg)	B—Condition of Substrate	C—Synthesis Time (min)	D—Transfer Voltage (V)	E—NaOH Concentration (M)
S_1_	22	Pristine	60	1	0.10
S_2_	22	Pristine	30	4	0.01
S_3_	22	Oxidized	60	1	0.01
S_4_	22	Oxidized	30	4	0.10
S_5_	11	Pristine	60	4	0.01
S_6_	11	Pristine	30	1	0.10
S_7_	11	Oxidized	60	4	0.10
S_8_	11	Oxidized	30	1	0.01

**Table 2 nanomaterials-13-02992-t002:** Alias structure of $2_{III}^{5-2}$ fractional factorial design and the relationship with $2^{3}$ full factorial design.

Parameters in Equation (1)	Effects Considered
$a_{0}$	I − ACD − BDE
$a_{1}$	A − CD + BCE
$a_{2}$	B − DE + ACE
$a_{3}$	C − AD + ABE
$a_{12}$	D − AC − BE
$a_{13}$	E − BD + ABC
$a_{23}$	AB + CE − ADE − BCD
$a_{123}$	AE + BC − ABD − CDE

**Table 3 nanomaterials-13-02992-t003:** 2^(5-2)^ matrix used for the fractional factorial design of the h-BN experiment. Average RMS roughness and contact angle responses as well as the RAMAN value of each of the samples are displayed.

Sample	A (mg)	B	C (min)	D (V)	E (M)	Roughness RMS (nm)	Contact Angle (°)	RAMAN (cm^−1^)
S_1_	22	Pristine	60	1	0.10	1.52 ± 0.21	75.0 ± 0.1	1366
S_2_	22	Pristine	30	4	0.01	1.35 ± 0.26	107.7 ± 1.1	1373
S_3_	22	Oxidized	60	1	0.01	2.66 ± 0.36	74.5 ± 1.9	1365
S_4_	22	Oxidized	30	4	0.10	1.05 ± 0.07	68.3 ± 4.4	1372
S_5_	11	Pristine	60	4	0.01	2.71 ± 0.53	74.5 ± 1.9	1363
S_6_	11	Pristine	30	1	0.10	1.19 ± 0.15	108.8 ± 0.5	1374
S_7_	11	Oxidized	60	4	0.10	1.92 ± 0.31	81.2 ± 0.6	1362
S_8_	11	Oxidized	30	1	0.01	1.75 ± 0.05	70.6 ± 0.4	1369

**Table 4 nanomaterials-13-02992-t004:** Analysis of variance of the regression model performed for the RMS roughness response.

Source	DF	Contribution	SC Ajust.	MC Ajust.	F-Value
Model	7	83.83%	16.78	2.40	29.61
Lineal	5	79.38%	15.89	3.18	39.26
A	1	3.56%	0.71	0.71	8.80
B	1	1.37%	0.27	0.28	3.40
C	1	45.08%	9.03	9.03	111.48
D	1	0.03%	0.01	0.01	0.06
E	1	29.34%	5.88	5.88	72.56
Two-term interactions	2	4.44%	0.89	0.44	5.50
A×B	1	4.41%	0.88	0.88	10.91
A×E	1	0.03%	0.01	0.01	0.08
Error	40	16.17%	3.24	0.08	
Total	47	100.00%			

**Table 5 nanomaterials-13-02992-t005:** Parameters to obtain maximum and minimum cases in roughness according to the model.

Case	A (mg)	B	C (min)	D (M)	Roughness RMS (nm)
Minimum	22	Pristine	30	0.10	0.63
Maximum	11	Pristine	60	0.01	2.73

**Table 6 nanomaterials-13-02992-t006:** Analysis of variance of the regression model performed for the contact angle response.

Source	DF	Contribution	SC Ajust.	MC Ajust.	F-Value
Model	7	98.96%	5559.13	794.16	217.25
Lineal	5	51.65%	2901.51	580.30	158.74
A	1	0.62%	34.67	34.67	9.48
B	1	34.00%	1909.99	1909.99	522.48
C	1	16.75%	940.98	940.98	257.41
D	1	0.05%	2.93	2.93	0.80
E	1	0.23%	12.95	12.95	3.54
Two-term interactions	2	47.31%	2657.63	1328.81	363.50
A*B	1	0.46%	25.72	25.72	7.04
A*E	1	46.85%	2631.91	2631.91	719.97
Error	16	1.04%	58.49	3.66	
Total	23	100.00%			

**Table 7 nanomaterials-13-02992-t007:** Parameters to obtain maximum and minimum cases in the contact angle according to the model.

Samples	A (mg)	B	C (min)	D (M)	Contact Angle (°)
Minimum	22	Oxidized	60	0.10	55.1
Maximum	11	Pristine	30	0.10	109.5

## Data Availability

Data are contained within the article and Appendix A.

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
