# Peer review of "Fractional Factorial Design to Evaluate the Synthesis and Electrochemical Transfer Parameters of h-BN Coatings"

_nanomaterials, 2023, doi:10.3390/nano13232992_

Round 1

Reviewer 1 Report

Comments and Suggestions for Authors

The authors proposed in this work a novel fractional analysis approach to explore the effects and interactions of key synthesis and electrochemical transfer parameters on the roughness and contact angle of two-dimensional hexagonal boron nitride coatings toward potential antibiofilm applications. The manuscript is well organized and structured, with enough modelling and experimental data supporting their key findings. I believe that this paper should be accepted after addressing the following concerns,

1. Why did the authors identify surface roughness and contact angle as critical factor on antibiofilm properties?

2.  The authors are suggested to include reference for the quantification of surface roughness.

3.   The quality of most figures should be further improved.

4.   For biofilm on antibacterial applications, the following references could be cited for a broader readership, e.g., 10.1016/j.carbpol.2023.120537; 10.1039/C8TB01245H.

Comments on the Quality of English Language

Minor editing of English language required.

Reviewer 2 Report

Comments and Suggestions for Authors

The article is devoted to determining the influence of synthesis parameters and subsequent processing on the properties of the h-BN films. To reduce the number of experiments, the authors use an unconventional approach to planning experiments, namely fractional factorial design. The article is interesting both from the point of view of the relevance of the material under study and the non-standard approach to optimizing the conditions for its formation. Generally, results of this article are good, which is deserved for our journal. There are some suggestions for the authors to improve their article.

1) The abstract is large in volume, most of it is a literature review. Please revise it and summarize all highlights of your research only.

2) The authors use a fractional factorial design. It is known that this method is effective in statistics.

But in materials science this method is not often used. Therefore, it is advisable to provide short information in the introduction on the possibilities and limitations of its use in materials science.

3) Line 100.

Add the chemical formula of the precursor.

4) Lines 101-102

If all copper substrates before deposition were oxidized and then etched in nitric acid, then the surface of both new and aged substrates should be the same. In this case, it is not clear to divide the substrates into “oxidized and pristine”.

That's why you couldn't detect the dependence on the type of substrate (Lines 385-389). I believe that the labeling of the substrates as “oxidized and pristine” are incorrect and is misleading to readers.

5) Lines 109-110

It is known that the key parameters of h-BN layer synthesis are deposition temperature and gas flow rates. Based on what, your preliminary research or literature data, these parameters were selected. Please explain or add references to the text of the article.

6) Lines 137-138. Please, add the conditions of oxidation the h-BN samples in air procedure as well as the results that it was oxidized. It is well known that when oxidized BN is destroyed and the oxidation product is B2O3 (white powder). It is not clear what led to “the improvement” in optical constants.

7) Line 175. “One sample was prepared for each experimental condition”

Surprisingly, only one sample was used for the bio-tests. Statistics are very important in these experiments.

8) Line 197. 

“that, according to what is reported in the literature, are determinants in the final quality of the substrate”.

Here you need to add references.

9) Line 243

The authors write about obtaining a monolayer of BN (please indicate its thickness). And at the same time, you got Rms=1-3 nm.

According to literature data, the BN monolayer has a significantly lower roughness.

Rms=0.269 nm (Nanotechnology 2015, 26, 275601); 0.4 nm (Sci Rep. 2016; 6: 30449); 0.2 nm (Journal of Crystal Growth 2016, 449, 148–155).

What is the roughness of your substrates? Add this value in the text. Maybe you have a large influence of the substrate?

10) Line 359-361

«an increase in precursor quantity is accompanied by a general decrease in roughness».

I cannot agree with this conclusion of the authors. The roughness value of samples C5 and C3, as well as C6 and C2, is the same.

11) Line 438

Please, explain why SiO2 was chosen as the comparison sample for bacterial tests?

12) Lines 445-447

Authors write that “A striking finding is connected to the robustness of this effect, because regardless of the hydrophilic or hydrophobic nature of the surfaces, we have observed no bacterial adhesion”.

As the authors indicated on line 175, they only used 1 sample for test the antibiofilm formation. Therefore, the conclusion is bold. You need statistics.

13) Line 450.

There is no description of figure 15.

14) There are typos in the text, for example:

- Line 244 - reference “602”

- Figure 14 is shown twice:  Lines 420-421 and Lines 430-431

- Line 471  “the roughness within the range of 0.63 to 2.73 μm”. What is the dimension of roughness values?

- Fig. S2  Optical microscopy of h-BN of h-BN samples ……

Reviewer 3 Report

Comments and Suggestions for Authors

The article may be interesting, but there are evident errors that need to be corrected first. It's possible that an incorrect version of the article was uploaded, or the authors did not thoroughly review the article before submission. In the PDF, almost half of the references are missing, and there are errors in the representation of references in the PDF file. Regarding the description of the factorial analysis in section 2.5, it's worth noting that some essential references necessary to explain the factorial analysis are missing and should be added.

A crucial condition that is not discussed in detail is the requirement to be in homoscedastic conditions. It's indeed not necessary to perform in triplicate, which corresponds to 11 experiments in the case of the selected factorial design (8 experiments plus 3 for 1 replication, again, considering  provided that the variance across the experimental domain is homogeneous.

Additionally, it is suggested to carefully check the tables, starting from the table that presents the model coefficients. There is an excessive number of significant figures, which is critical in an article that discusses the use of experimental design. This issue is also noticed in other tables, such as the one reporting contact angle data, where there is no mention of standard deviation, despite its presence in the graphs.

Particular attention should be paid to the labels in the figures. In some cases, the original language of the author has been left, which should be standardized. Furthermore, the division of samples in Figure 10 and subsequent figures is not easily interpretable. Although the precursor content is reported first, there is no clear order either by concentration or by the sample number. Please ensure uniformity in this regard.

Aside from these obvious and serious errors, it is suggested to reduce the length of the article, for example, by shortening the section describing the experimental setup. Nevertheless, the article may be of interest to readers, and after careful revision, publication is recommended.

Comments on the Quality of English Language

I think the quality of English is good

Reviewer 4 Report

Comments and Suggestions for Authors

The authors study the effect of different experimental condition on surfaces properties of h-BN coatings. Through fractional factorial, statistical analysis (ANOVA), they tried to understand the influence of different experimental variables (e.g. solvent concentration, and so on) on h-BN coating. The aim is to obtain a large amount of information from a limited number of experiments. They also made an investigation to understand an antibiofilm properties of these coatings. The topic is of interest, but I doubt the paper is acceptable in its current form.

In general, I have doubts about the chosen evaluation method to determine whether the coatings have an effect from a scientific perspective. They grew bacterial suspensions of E. Coli under non-optimal conditions (30 °C, typically 37 °C is suggested in the literature) and then suspended them on a PBS saline buffer, applying a single drop of these suspensions to the samples for 24 hours. They simply place drops of 300 μL on samples. Under these conditions, I don't believe the bacteria have either the time or the nutrients to grow and form biofilm. Furthermore solvent evaporation might cause osmotic stress to the cells. This test could only eventually assess potential adhesion in an adverse environment. Furthermore, SEM images (Fig. 15) provided for this test (the only result of this presumed anti-biofilm activity) have incorrect scale bars (see comment n.7). Even a simple crystal violet test could provide more meaningful insights. Furthermore, the quality of the figures in the text is questionable and, in general, not good. 

Some example are below.

1.     in some graphs, the units of measurement are presented in round brackets, while in others, they are in square brackets,

2.     in Figure 5 scale bars are missing,

3.     in Figures 7 and 9, the axes are not in English,

4.     Some similar graphs exhibit different quality or fonts (e.g., Figure 13 has lower quality compared to Figure 14,

5.     line 421 presents a figure without a caption,

6.     in Figure 14 the textures chosen for the voltages are indistinguishable, making it impossible to read,

7.     in Figure 15 the scale bars are evidently incorrect, and in C and D, I doubt the scaling is accurate (just knowing bacteria dimension is possible to understand that those scale bar are from larger images),

8.     in Line 156 and 157, it would be helpful to specify the camera model and add reference to the ImageJ plugin used,

9.     in line 164, specify the components of the growth medium,

10.  in line 239, there is a message of error, maybe due to bibliography program used. There are recurring errors of this type in the text, for brevity, I will only mention this one.

11.  in line 291, Table 5 has a different graphic style compared to other tables. I recommend standardizing them with a single graphic style.

12.  in lines 436-438, be cautious with these claims; additional experimental evidence would be needed to support them.
